# A Virtuous Circle? Increasing Local Benefits from Ports by Adopting Circular Economy Principles

**Toby Roberts** [1,*] , **Ian Williams** [1] , **John Preston** [2] , **Nick Clarke** [3] , **Melinda Odum** [3] and **Stefanie O'Gorman** [4]

1 Infrastructure Group, Faculty of Engineering and Physical Sciences, University of Southampton, Southampton SO17 1BJ, UK; idw@soton.ac.uk

2 Transportation Group, Faculty of Engineering and Physical Sciences, University of Southampton, Southampton SO17 1BJ, UK; J.M.Preston@soton.ac.uk

3 Ramboll UK Ltd., Southampton SO40 7HT, UK; Nick.Clarke@Ramboll.co.uk (N.C.); Melinda.Odum@Ramboll.co.uk (M.O.)

4 Ramboll UK Ltd., Edinburgh EH2 3AH, UK; stefanie.ogorman@ramboll.co.uk

* Correspondence: t.j.roberts@soton.ac.uk

**Abstract:** As ports seek to maintain support for their operations amidst growing environmental awareness and social pressure, it is important they provide benefits for the local population to offset negative impacts. Ports can add additional economic benefits for the cities they are located in by encouraging maritime clusters, industrial development, a circular economy, and waterfront development. The current level of adoption, interest in future adoption, barriers to implementation, and attitudes towards the views of the local population were assessed via an online questionnaire sent to port authorities in 26 countries. The potential and willingness of ports to be on the frontline of the transition to a circular economy globally has been clearly identified for the first time, seeing a 60% increase between current levels of adoption and future interest in adoption. Barriers to a circular economy are comparable to barriers to widely adopted methods, such as industrial development and a waterfront economy. It is likely that circular economy activities in port cities will add additional local benefits and reduce the negative impacts of a port. A new framework is proposed to help ports and cities collaborate and encourage greater adoption of the circular economy.

**Keywords:** port cities; maritime clusters; industrial development; circular economy; waterfront development; sustainable development; economic; benefits; framework

## 1. Introduction

Ports were developed to facilitate the movement of goods and people; however, they provide numerous other benefits. These range from the obvious benefits provided by port activity, such as employment, to less obvious benefits, such as knowledge spillovers created by the pooling of people and industries in one place. However, as the adverse impacts of ports' and the awareness of these impacts have grown, the activities of ports have increasingly been a source of concern [1]. Whilst the importance of ports for national economies and global trade is clear, there has been a decline in port-related benefits at a local level [2]. This has occurred due to increasing adverse environmental impacts, awareness of these impacts, relocation of port-related activity, decreasing employment and the casualisation of that employment, and the increasing use of international (rather than local) inputs. These changes have been driven by globalisation and technological innovations, such as growing ship sizes, mechanisation, and the rise of containerisation, with containerisation dramatically reducing the number of people that need to be employed directly in the port [3]. Additionally, the levels of economic, institutional, and infrastructural integration between ports and cities have decreased, with ports serving both consumers and producers over much wider geographical areas and hinterlands [4]. The OECD [5] found that 90% of the economic benefits of ports now occur outside of the port city area and Jung [6]

concluded that local economies might no longer significantly benefit from the existence of nearby ports.

If ports are to operate more sustainably and in greater harmony with their local areas, this issue needs to be addressed. Improving relationships with the local community has become a high priority for ports, with the European Sea Ports Organisation listing improving relations as number 5 on their top 10 environmental priorities list [7]. Increasing the local economic benefits from ports is one way to address this. If port cities of the future are to be sustainable, ports must aim to reduce their negative environmental impacts as much as possible, whilst increasing their local economic and social benefits. A greater understanding of the ways ports can increase local benefits, the levels of adoption of these approaches globally, and the barriers ports have to overcome to implement them are needed.

Historically, industrial development, waterfront development, and the development of maritime clusters have been regarded as key ways for ports to add economic value. As sustainable development has become a key focus of research, the circular economy (CE) has received increasing attention as an additional means of adding value, whilst reducing environmental impact and increasing social equity. The circular economy is defined by Kirchher et al. [8] as being "an economic system that replaces the 'end-of-life' concept with reducing, alternatively reusing, recycling and recovering materials in production/distribution and consumption processes. It operates at the micro-level (products, companies, consumers), meso-level (eco-industrial parks) and macro-level (city, region, nation and beyond), with the aim to accomplish sustainable development, thus simultaneously creating environmental quality, economic prosperity and social equity, to the benefit of current and future generations." This allows waste to be regarded as resources, which can be recycled and reused [9]. The circular economy allows ports to become more sustainable whilst creating value in the process. Despite the considerable potential of the circular economy, implementation is not widespread [9], and more research is needed to help increase implementation. This is especially true of ports, which have only emerged as research interests—in relation to the circular economy—in recent years [10]. The circular economy may also allow industrial development, waterfront development, and maritime clusters to amplify their local benefits and reduce negative impacts.

### 1.1. The Economic Value of Ports

The benefits of ports are clear and numerous. They are key drivers of economic development in the local area and hinterland, by handling over 80% of world trade [11], facilitating the movement of people, creating infrastructure development, supporting direct and indirect job provision, attracting investment, and lowering costs for producers and consumers. This can be illustrated using the example of the UK, where ports handle 486 million tonnes per year [12], 64.5 million passengers [13], support GBP 70 billion in turnover, and 822,000 jobs [14]. However, different port-related activities bring different levels of economic benefits and employment opportunities for the local population. A good example is that automobiles as cargo adds USD 220 of value added per metric tonne on average, compared to just USD 20 for grain [5]. Therefore, the function of a port affects the level of benefits it provides for its local economy. The presence, or lack, of additional port-related services and activities, such as ship registry, industrial development, and ship repair, can affect the economic value a port provides for a city. As an example of the benefits a port can provide, the port of Southampton directly (or indirectly) supports up to 10% of the city's employment [15], supports 45,600 jobs, and contributes GBP 2.5 billion to the national economy [16]. However, the benefits of ports' economic activities are becoming increasingly wide spread, whilst the negative externalities created by ports remain concentrated in the local area. Again, for Southampton, many of the jobs supported by the port are located outside of the local area, such as 11,700 jobs in the automotive sector, nearly 200 km away in the west midlands [16]. Some practices in ports, such as transhipment, provide little value to the local area. A good example is found in Rotterdam, which is able to host the

largest ships arriving from other continents, and move cargo onto smaller vessels before shipping them on to UK ports [17]. This allows the port to increase its competitiveness, but provides few benefits to local people. Due to the economic benefits of ports spreading more widely, the city may wish to increase the economic value it creates for the local area.

Geerlings et al. [18] and OECD [5] illustrate three key ways for ports to add economic value to cities: waterfront economies/development, maritime clusters, and port-industrial development. These three measures are well researched and established in many port cities; however, the circular economy is also receiving increasing attention as a new potential way for ports to add additional local benefits, whilst reducing their environmental impact. Encouraging these approaches enables a port to augment value beyond the simple movement of goods and people. There is a lack of research investigating all of these approaches together on a global scale, with research focusing mostly on the adoption or barriers to one of these approaches and mostly focusing on case study cities or regions.

### 1.2. Circular Economy

Despite growing discussion and focus on the circular economy (CE), Kirchherr et al. [8] identified 95 different definitions of the circular economy in the literature, highlighting that there is still a lack of consensus over the exact definition. The definition provided by Kirchherr et al. [8] attempts to provide a universal definition encompassing all the key aspects identified in the literature. Therefore, this definition will be used for this article. The differences between the circular economy and the linear economy are shown in Figure 1.

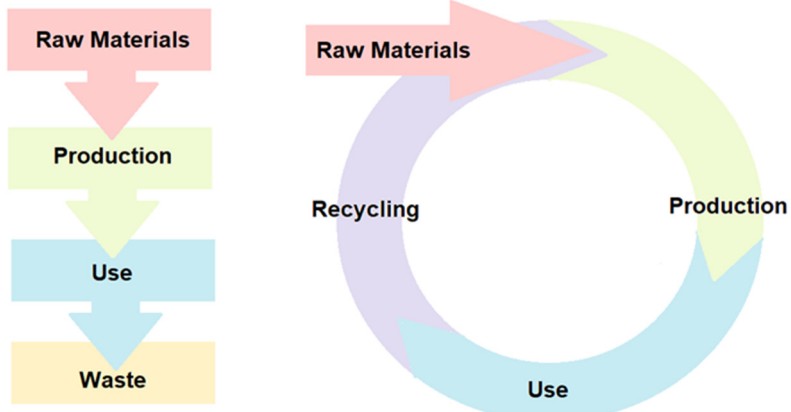

**Figure 1.** Linear economy (**left**), compared to a fully circular economy (**right**), adapted from Bianchini et al. [19] (Copyright 2019 Published with CC-BY License).

The CE is regarded as an approach, which decouples economic growth from environmental impact [20], and it achieves this by the implementation of 10 key circular economy principles (10 Rs). These are refuse, rethink, reduce, reuse, repair, refurbish, remanufacture, repurpose, recycle, and recover [21]. This approach focuses on every stage of a product's life cycle creating considerable opportunities and potential benefits. This can not only reduce environmental impacts, but also improve raw material's supply security, increase economic growth, improve the durability and lifetime of products, stimulate innovation, and increase competitiveness [22]. The circular economy also has the potential to create considerably more jobs, with Gaia [23] finding that reuse creates 200 times more jobs than landfills and incinerators. The circular economy has the potential to provide a vast array of economic, social, and environmental benefits, enabling greater levels of sustainable development.

Despite its considerable potential, the circular economy faces numerous barriers to implementation. Kirchherr et al. [24] identified the main barrier to circular economy to be cultural attitudes, such as a lack of consumer interest and awareness, and a lack of knowledge and collaboration between businesses and stakeholders [25]. Other key barriers

identified in the literature are a lack of policy support [26], lack of a consistent framework [25], technological limitations [27], and a (perceived) lack of financial viability for circular economy business models [5]. Van den Berghe et al. [28] highlighted space and land-use conflicts between expanding circular economy industries and expanding residential areas in cities as a potential source of tension. Low prices of many raw materials mean that reusing waste products may be less financially viable than using raw materials [25,29]. High investment costs are a barrier to many circular economy ideas that require new infrastructure [29]. If the circular economy is to be implemented, these barriers will need to be better understood and overcome. The circular economy has also been described as a niche discussion, lacking wider acceptance and acknowledgement [24], which places it at risk of failing to achieve widespread adoption.

In order for these barriers to be overcome, key enabling factors can be encouraged to create better conditions for CE to thrive [25]. Hart et al. [25] identified cultural, regulatory, and financial enablers, which can help overcome the challenges facing the circular economy. Among these, leadership has been highlighted as a key enabler for CE, with leadership from the top being seen as critical [25]. This is further demonstrated by Moktadir et al. [30], who highlighted the importance of leadership on CE. Cooperation and networking within and between businesses and stakeholders has also been highlighted as a key enabler of CE [10,31–33]. Another key factor that emerges from the literature is the importance of information sharing and awareness of CE [30,32]. Regulatory enablers, such as policy support, regulatory reform, and incentives for CE are also important [25]. Creating a strong business case for CE can also be used to overcome the financial barriers [25,32].

Circular Economy and Port Cities

Gravagnuolo et al. [31] stated that the transition to a circular economy requires a cultural paradigm shift across all sectors. Port cities may be ideally placed to lead this transition due to being a hotbed of industry, end-of-life products, and secondary raw materials, as well as potential consumers [34]. Due to this, CE in ports has become an emerging area of research in recent years [10].

Ports adopting CE can help add economic value, reduce the environmental impacts, and provide social benefits, making it an excellent example of sustainable development. The circular economy transition in ports has the potential to create employment opportunities in a range of new sectors depending on the opportunities available in each port. These jobs can occupy all skill levels within the labour market, and be used to encourage development in deprived regions [35]. This can help increase local support for the port and contribute to the social license to operate, addressing some of the issues created by declining port employment due to trends such as increasing automation. This addresses the often-neglected social dimension of sustainability.

ESPO [7] discovered that ship waste and port waste are the sixth and eighth highest ranked environmental priorities of European ports, highlighting the importance of this issue to port authorities. Examples of the circular economy within ports are mostly found within Europe and North America [18]. Considering that many of the world's largest and most impactful ports are found in the developing world [36], greater emphasis must be placed on the circular economy globally and greater research is needed in these regions.

Some examples of a circular economy in ports are already adopted, such as reusing dredged materials. This can be seen in the port of Gavle, Sweden, where dredged material was used to create new land for port expansion [37]. This enabled the port to grow whilst reusing waste produced via routine dredging in the port. A further example can be found in the port of Southampton, which uses dredged materials for a range of beneficial purposes, such as beach replenishment [38]. The port of Rotterdam captures $CO_2$ from industrial processes within the port area and provides it to local greenhouses to improve crops' growth [39]. Antwerp's port reuses waste products from the petrochemical industry that can be reused by other companies, benefiting from economies of scope [40]. This suggests that large ports may be well situated to utilise circular economy principles.

Companies such as Qpinch have demonstrated ways to reclaim residual heat from the port whilst simultaneously reducing energy bills [41] Old port land can be repurposed for new purposes and benefit from existing facilities. An example of this, which has already been implemented in many ports, is combined heat and power schemes (CHP), such as the residual waste incinerator in Plymouth, UK, which provides hot water to residential areas within the naval base via existing naval piping systems [42] or the Port of Liverpool's CHP plant [43]. There is also considerable potential for circular economy thinking to be applied to issues that are unique to ports and have been poorly addressed to date. A good example of this is the correct disposal—and reuse or recycling—of end-of-life fishing gear and ropes. Abandoned, lost, or otherwise discarded fishing gear creates a range of environmental, as well as socioeconomic impacts [44]. Correctly disposing of, repairing, and re-using fishing gear would contribute greatly to reducing these impacts. This will require the facilities to receive, collect, process, recycle, and reuse fishing equipment. It may be difficult for smaller ports to build this infrastructure, which provides an opportunity for larger ports with the required infrastructure to provide a service for their own vessels, as well as those in surrounding ports that lack such infrastructure. A further example of an issue that would benefit from a circular economy (from ports) is the handling of sludge, waste, and wash water from open and closed-loop scrubbers. Wash water from open loop scrubbers disposed of at sea has been shown to contribute to acidification, as well as polluting the environment with polycyclic aromatic hydrocarbons, heavy metals, nitrogen, and SOx [45]. Closed-loop scrubbers should therefore be the preferred option; however, ports will need the facilities to deal with this. Whilst it remains cost-effective to utilise scrubbers rather than switching to low sulphur fuel, these issues can be expected to become more widespread. It is essential that ports of the future are equipped to deal with this waste and reuse/recycle it wherever possible. For example, sludge from scrubbers may have other uses, such as in cement [46].

Port cities experience many of the same enabling factors and barriers to CE as non-port cities; however, there are some challenges that may be unique to the port area, such as the wide array of potential stakeholders. This issue is highlighted in the literature, with Gravagnuolo et al. [31] stating the importance of cooperation in port cities if CE is to be implemented, and Mankowska et al. [10] identified the importance of communication between port authorities and external stakeholders. Haezendonck and Van den Berghe [20] state that although port authorities must play a key role in CE implementation within ports, networking, exchange of ideas, and funding provision is crucial. Girard [47] illustrated the importance of collaboration with stakeholders from outside the port area, eventually enabling CE over larger areas, growing from industrial symbiosis within the port, to urban symbiosis within the port city, and eventually city-territorial symbiosis, including the wider area and hinterland. Port cities, especially when the port is privatized, may suffer from a lack of unified leadership, due to the competing interests of port and city authorities. This makes implementing CE potentially more challenging.

Ports face a variety of challenges when implementing CE, such as transportation and infrastructural issues, availability of suitable supply chain partners, product traceability, uncertainty of return, and high up-front costs [10]. Key cultural barriers, such as resistance to change, coordination, and information sharing have been identified [10], as well as the challenges presented by varying types of ports all having unique opportunities and challenges. This makes creating a universal framework for port cities more challenging. Overall, these is a lack of research on the barriers to CE with a focus on port cities.

Although research on CE has increased in recent years, there is a lack of research focusing on port cities, especially those using surveys and questionnaire data [48] rather than literature reviews, and a lack of global research, with most work focusing on case study cities or regions, mostly in developed countries. This is especially true for port cities, with Zheng et al. [49] illustrating how there is a greater need for work with a more global focus, especially that which incorporates examples from Asia, Africa, and South America. The views of port authorities globally on circular economy are unclear due to the fact

certain regions have been neglected in the research. Mankowska et al. [10] also highlights the need for greater work, focusing on secondary ports rather than only focusing on the world's largest ports.

### 1.3. Industrial Development

Alongside CE, more traditional ways for increasing the economic benefits of ports must also be considered as an important means of adding economic benefits. Despite the negative impacts of the industry, such as air pollution, industrial development in a port has been an essential source of economic benefits. Historically, port development and industrial development have taken place hand-in-hand [18], due to being a break in bulk point, where cargo is transferred from one form of transport to another. This led to the growth of supplementary activities and industries around bulk points, such as manufacturing [50]. Despite the negative environmental impacts, the economic benefits of industrial development are clear and CE principles could be used to enable industrial development to take place whilst reducing the environmental impact. This is especially important in developing countries where industrial development may be pursued as a path to growth. Policies can be implemented to encourage industrial development within port areas, such as special economic zones, as in Shenzhen [51]. Ports, such as Shenzhen, have been able to benefit from the creation of a technopole, allowing the port's growth to be fuelled by export-led high-tech manufacturing. Port cities that contain technopoles may experience rapid technological and economic growth by benefitting from agglomeration effects. High tech manufacturing and technopoles in and around ports may become more common in the future as the quaternary sector continues to expand, making up a larger percentage of port industrial development. If this is the case, it is particularly important that these areas pursue CE principles to mitigate the impacts whilst providing greater benefits.

Globally, there has been a shift, with industry moving from developed nations to developing nations since the second half of the 20th century [52]. This, in many cases, has led to a decline in port industry in developed countries and an expansion in developing countries. In places like China, industrial development within a port may be encouraged through measures, such as special economic zones [53]. However, in many developed countries, these facilities are increasingly under pressure due to their environmental impact and the pursuit of greener paths to growth [54]. Despite this, it has been suggested that free ports or special economic zones could be implemented in the UK post-Brexit to encourage economic growth in deprived areas [55].

Free ports allow the movement of goods into the free port without paying tariffs, which can then be processed into final goods and exported without paying tariffs, or sold in the domestic market once the tariffs are paid on the final goods [55]. There may also be additional benefits due to the agglomeration effect, where the close proximity of related businesses leads to an economic advantage. It is hoped that this will spur economic growth in these areas, with Mace [56] suggesting free ports in the UK could lead to 150,000 new jobs and contribute GBP 9 billion annually to the national economy. It is debatable as to whether or not free ports actually create new businesses or employment, or simply lead to a relocation of existing businesses and employment to the free port area [57]. It is therefore unclear whether this policy will actually create additional local benefits, with Serwicka and Holmes [57] concluding that the net benefit to the UK economy would be small. Free ports may however be used to encourage new and more sustainable industries, such as developing hydrogen hubs, such as the proposed Freeport East Hydrogen Hub [58].

Industrial development in port areas may be inevitable, and one possible way for ports to benefit from industrial development within the port, whilst reducing the negative impacts on the city, is through circular economy principles.

### 1.4. Waterfront Economy and Regeneration

Waterfront regeneration and development can also create numerous social, economic, and environmental benefits. As ships became larger and more advanced, older port

areas have increasingly fallen into decline if they are unable to be adapted to fit the new requirements [59]. This has left many cities with areas of abandoned inner-city port land. In many cases, the reuse of this port land has boosted the local economy and created new areas and facilities that benefit the local communities. Hoyle [60] highlights how decline in inner-city port areas has led to waterfront development and urban renewal, leading to revitalised inner cities. Alongside this, an increasing focus on the quality of life and pollution reduction in port cities has led to many particularly polluting industries being moved from the traditional port areas at the heart of cities to less central locations [61]. A good example of waterfront regeneration can be found in Baltimore, which was among the earliest to regenerate its waterfront in 1963. This allowed Baltimore to revitalise a degraded area of port land and set an example, which many port cities worldwide have subsequently tried to emulate, with varying degrees of success [62]. Some examples of waterfront development include Port Vell, Barcelona [63], HafenCity, Hamburg [64], Royal William Yard, Plymouth [65], Red Brick Warehouse, Yokohama [66], and Inner Harbor, Baltimore [67]. Regeneration can include a range of commercial, residential, tourism and leisure facilities. Although all ports would hope to remain competitive, inevitably, some will not. Even within successful ports, areas of port land may no longer be required. Cities often wish to have greater access to the waterfront and the economic advantages that it brings. Waterfront regeneration is an effective way for a city to benefit from its waterfront, whilst in many cases maintaining its maritime heritage. CE principles could also be applied to this redevelopment and reuse of land, creating less impactful development.

### 1.5. Maritime Clusters

A further effective way for a port to provide additional localised economic benefits is to develop a maritime cluster. A maritime cluster refers to an agglomeration of interconnected port-related industries within a port city [68], such as ship building, coastal tourism, maritime services, fisheries, dredging and shipping amongst others [69]; see Figure 2.

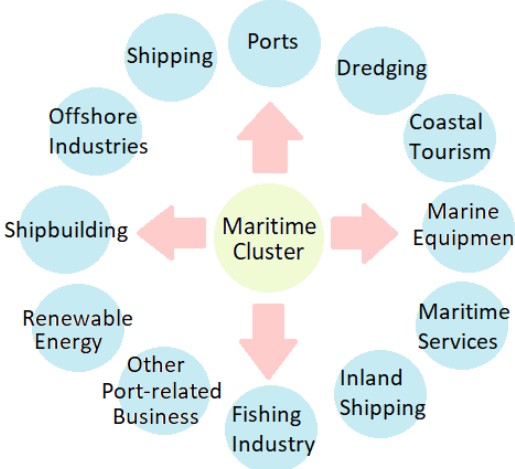

**Figure 2.** Example of maritime cluster composition.

Maritime clusters help increase the port's economic benefit, whilst strengthening the port's status and providing added resilience if the port activity were to decline. Li and Luo [70] described a maritime cluster as an ecosystem in itself, where maritime-related industries and institutions can work and develop together in a symbiotic way. This provides benefits for the businesses involved in the cluster and the local and national economies. A good example of this is London, which, despite being behind other UK ports, such as Southampton, in terms of cargo tonnage and passenger numbers, possesses a maritime cluster that is considered to be among the most extensive in the world [18]. This may be due to the path dependency effect and the fact London's historic status as a major global port has led to it developing such a well-established maritime cluster [71]. The example

of London suggests a maritime cluster can provide longevity as a source of economic benefits. The formation of maritime clusters benefits both the port and the city, and ports and cities can work together to encourage their growth. This symbiotic system may provide a network for CE activities to take place, taking advantage of the existing connections and coordination between stakeholders.

*1.6. Research Gap*

There is a lack of research investigating the levels of interest, levels of adoption of, and barriers to, a circular economy in port cities on a global scale, considering ports of all sizes and ports from countries at all stages of development. This paper aims to identify the global potential of the circular economy within ports, by investigating the levels of adoption, interest and key barriers, with comparisons to maritime clusters, industrial development, and waterfront development. This paper will also propose a framework to encourage greater levels of circular economy activity in port cities.

## 2. Materials and Methods

An online questionnaire was distributed to professionals employed by port authorities in 26 countries (Albania, Australia, Belgium, Belize, Brazil, Egypt, Canada, Chile, China, Finland, France, Japan, Latvia, Morocco, Namibia, Netherlands, Portugal, Romania, Saudi Arabia, Singapore, South Korea, Spain, Sri Lanka, Togo, UK, and the USA). Response levels varied from a maximum of 51 to a minimum of 33 respondents per question. This was an acceptable number of participants for research into ports and compares with the sample size of other similar studies, such as Moeremans and Dooms [72], and ensured the sample was global, addressing the call for greater representation from developing regions [48].

Ports were initially targeted using lists of the world's largest ports according to cargo tonnage [73], TEU [74] and passenger numbers [75–77]. Additional ports were included if relevant professionals with suitable expertise were found during the search process, regardless of their size. The final sample contains 16% of the world's top 100 container ports [74], 10% of the world's top 100 largest cruise ports [77], and a geographical distribution covering a large variety of countries at varying levels of development, as well as including smaller ports that have been left out of previous studies [10].

Professionals with adequate expertise in port authorities were identified in numerous ways, such as using port authority websites and the professional networking website LinkedIn. In addition, the British Ports Association (BPA) distributed the questionnaire to UK ports. Participants linked to a port's LinkedIn page were approached if their role in the port was related to management, operations, planning, engineering, or the environment. In some cases, the final participant was recruited via recommendations from the person who was contacted initially. The questionnaire contained political, economic, social, technological, environmental, and legal (PESTLE) sections. The PESTLE framework was chosen as an appropriate tool for conducting a broad fact finding exercise and has been shown to be useful in analysing both internal and external factors [78], which is highly important for work on ports.

Current levels of adoption and future interest in encouraging maritime clusters, industrial development, circular economy, and waterfront development were all recorded as yes or no responses. These were analysed using chi-squared and $2 \times 2$ contingency tables. The chi-squared based measure of association Phi coefficient was used as a measure of association due to the low sample size and occasional cells with a count of less than 5 [79]. A phi of >0.5 was considered a high association, 0.3–0.5 medium association and 0.1–0.3 as a low association. Yates' correction was used to provide continuity corrected chi-squared values and *p* values. Yates' correction effectiveness is disputed, with some sources saying it tends to be too conservative [80]; therefore, Fisher's exact test was also provided as a further level of scrutiny. Spearman's rank correlation was used to analyse the port size and GDP per capita of host countries with responses. Spearman's rank was chosen due

since the responses are measured with categorical responses (yes or no) or an ordinal Likert scale, making the use of non-parametric statistics the preferred option [80].

The size of obstacles to maritime clusters, industrial development, circular economy, and waterfront economy were recorded using a Likert scale (very small, small, medium, large, and very large). This data were further investigated by creating mean and median values and ranking the barriers. Mean and median were chosen as the most appropriate measures of central tendency for Likert scales [81].

The answers of the respondents to the following questions/statements were also analysed:

Please rate your level of agreement with the following statements:

- It is important for the local population to be knowledgeable about the port.
- It is important for the port to create benefits for the local population.
- Do you feel the local population is aware of the benefits the port provides?
- Do you feel the attitude of the local population towards the port is positive?
- How interested is the port in improving the attitude of the local population towards the port?
- Does the port feel under pressure from local residents to reduce its negative impact?

These responses were recorded on a Likert scale (1–5); however, responses were converted to a simple yes or no to make analysis using chi-squared possible by increasing the cell counts. The mid-point on the Likert scale was categorised as a no as it showed a lack of agreement with the statements, so this analysis focused on identifying the frequency of positive responses. This was then analysed using chi-squared following the same method outlined earlier. Statistical analysis was undertaken using Statistical Package for the Social Sciences (SPSS) software [82], and the findings, which were significant at the 95% confidence level, are presented.

## 3. Results

The participants were involved in various areas of port authority operations, such as management (37.2%), planning (30.2%), environment (11.6%), engineering (4.6%), commercial (6.9%), administration (4.6%), and others (4.6%). Using the Southampton System's port grouping as a framework [83], this survey included four micro-ports, nine small-ports, 12 medium-ports, 13 large-ports, and 13 ports that chose not to specify their size, some of which were possible to group from publically available online data for cargo and passenger numbers of that port.

The current levels of adoption of maritime clusters, port industrial development, circular economy, and waterfront development, as well as the level of interest in encouraging them in the future, are presented in Table 1. This shows that, of the four methods considered in this study, the circular economy is the method with the lowest reported current levels of adoption, and it is the only option described as showing a large increase between current levels and future interest (58.8%). Maritime clusters and port industrial development are relatively stable between current levels and future interest; however, waterfront economy shows a fall in interest from current levels.

The levels of association, in regards to the presence of maritime clusters, port industrial development, circular economy, and waterfront development are shown in Table 2. Continuity correction was used when required, however sections where it was not necessary or not possible are left blank. This is also the case for later tables. Table 2 shows a statistically significant association among the current adoptions of all methods considered in the study, with the adoption of one method increasing the likelihood of adopting another. The strongest associations among these are with waterfront development and the presence of circular economy and industrial developments.

**Table 1.** Current presence (and levels) of interest in encouraging various methods of increasing the economic benefits of ports (%).

|  | Present | Not Present | Interested in Encouraging in the Future | Percentage Change |
|---|---|---|---|---|
| Maritime clusters | 23 | 28 | 23 | 0 |
| Port industrial development | 30 | 19 | 31 | 3.3 |
| Circular economy | 17 | 29 | 27 | 58.8 |
| Waterfront economy | 33 | 18 | 24 | −27.2 |
| None | 5 | NA | 2 | −60 |

**Table 2.** The presence of a method for increasing a port's economic benefits and the presence of other methods.

|  | Value | Presence of a Maritime Cluster | Presence of Port Industrial Development | Presence of Circular Economy | Presence of Waterfront Development |
|---|---|---|---|---|---|
| Presence of a maritime cluster | Chi-squared continuity correction Phi | NA | 6.117 4.652 0.373 | 9.031 0.453 | 5.107 3.768 0.341 |
|  | Sig (two-tailed) continuity correction | NA | 0.013 0.031 | 0.003 | 0.024 0.052 |
|  | N | NA | 44 | 44 | 44 |
| Presence of port industrial development | Chi-squared continuity correction Phi | 6.117 4.652 0.373 | NA | 8.264 6.521 0.433 | 10.471 0.488 |
|  | Sig (two-tailed) continuity correction | 0.013 0.031 | NA | 0.004 0.011 | 0.001 |
|  | N | 44 | NA | 44 | 44 |
| Presence of circular economy | Chi-Squared continuity correction Phi | 9.031 0.453 | 8.264 6.521 0.433 | NA | 11.039 8.995 0.501 |
|  | Sig (two-tailed) continuity correction | 0.003 | 0.004 0.011 | NA | 0.001 0.003 |
|  | N | 44 | 44 | NA | 44 |
| Presence of waterfront development | Chi-squared continuity correction Phi | 5.107 3.768 0.341 | 10.471 0.488 | 11.039 8.995 0.501 | NA |
|  | Sig (two-tailed) continuity correction | 0.024 0.052 | 0.001 | 0.001 0.003 | NA |
|  | N | 44 | 44 | 44 | NA |

The levels of association, in regards to the presence of maritime clusters, port industrial development, circular economy, and waterfront development, and interest in encouraging these methods in the future, are presented in Table 3. It shows levels of association among the reported current presence of the four methods and levels of interest in encouraging them in the future. The presence of a maritime cluster is associated with interest in maritime clusters and port industrial development. The presence of port industrial development is associated with interest in port industrial development, circular economy, and waterfront

developments. The presence of existing circular economy is associated with interest in encouraging port industrial development and circular economy. The presence of waterfront development is associated with interest in port industrial development, circular economy, and waterfront development.

**Table 3.** The presence of a method for increasing a port's economic benefits and levels of interest in encouraging the other methods.

| | Value | Interest in Encouraging a Maritime Cluster | Interest in Encouraging Port Industrial Development | Interest in Encouraging Circular Economy | Interest in Encouraging Waterfront Development |
|---|---|---|---|---|---|
| Presence of a maritime cluster | Chi-squared continuity correction | 9.031 | 7.232<br>5.638 | 2.199 | 2.448 |
| | Phi | 0.453 | 0.405 | 0.224 | 0.236 |
| | Sig (two-tailed) continuity correction | 0.003 | 0.007<br>0.018 | 0.138 | 0.118 |
| | N | 44 | 44 | 44 | 44 |
| Presence of port industrial development | Chi-squared continuity correction | 2.530<br>1.612 | 10.748 | 6.145 | 8.729<br>6.996 |
| | Phi | 0.240 | 0.494 | 0.374 | 0.445 |
| | Sig (two-tailed) continuity correction | 0.112<br>0.204 | 0.001 | 0.013 | 0.003<br>0.008 |
| | N | 44 | 44 | 44 | 44 |
| Presence of circular economy | Chi-squared continuity correction | 0.354 | 5.948<br>4.492 | 5.981<br>4.525 | 1.453 |
| | Phi | 0.090 | 0.368 | 0.369 | 0.182 |
| | Sig (two-tailed) continuity correction | 0.552 | 0.015<br>0.034 | 0.014<br>0.033 | 0.228 |
| | N | 44 | 44 | 44 | 44 |
| Presence of waterfront development | Chi-squared continuity correction | 0.540 | 3.012 | 7.943<br>6.307 | 11.189<br>9.200 |
| | Phi | 0.111 | 0.262 | 0.425 | 0.504 |
| | Sig (two-tailed) continuity correction | 0.462 | 0.083 | 0.005<br>0.012 | 0.001<br>0.002 |
| | N | 44 | 44 | 44 | 44 |

The levels of association, in regards to the interest in encouraging maritime clusters, port industrial development, circular economy, and waterfront development are shown in Table 4. It shows that interest in maritime cluster development is associated with interest in port industrial development and circular economy. Interest in port industrial development is associated with interest in maritime cluster development, circular economy, and waterfront development. Interest in the circular economy is associated with maritime cluster development and port industrial development. Interest in waterfront development is associated with interest in port industrial development only.

**Table 4.** The interests in encouraging a method for increasing a port's economic benefits and interests in other methods.

| | Value | Interest in Encouraging a Maritime Cluster | Interest in Encouraging Port Industrial Development | Interest in Encouraging Circular Economy | Interest in Encouraging Waterfront Development |
|---|---|---|---|---|---|
| Interest in encouraging a maritime cluster | Chi-squared continuity correction Phi | NA | 5.948 4.492 0.368 | 9.501 7.640 0.465 | 1.453 0.182 |
| | Sig (two-tailed) continuity correction | NA | 0.015 0.034 | 0.002 0.006 | 0.228 |
| | N | NA | 44 | 44 | 44 |
| Interest in encouraging port industrial development | Chi-squared continuity correction Phi | 5.948 4.492 0.368 | NA | 4.619 0.324 | 14.491 12.242 0.574 |
| | Sig (two-tailed) continuity correction | 0.015 0.034 | NA | 0.032 | 0.000 0.000 |
| | N | 44 | NA | 44 | 44 |
| Interest in encouraging circular economy | Chi-squared continuity correction Phi | 9.501 7.640 0.465 | 4.619 0.324 | NA | 0.748 0.130 |
| | Sig (two-tailed) Continuity correction | 0.002 0.006 | 0.032 | NA | 0.387 |
| | N | 44 | 44 | NA | 44 |
| Interest in encouraging waterfront development | Chi-squared continuity correction Phi | 1.453 0.182 | 14.491 12.242 0.574 | 0.748 0.130 | NA |
| | Sig (two-tailed) Continuity correction | 0.228 | 0.000 0.000 | 0.387 | NA |
| | N | 44 | 44 | 44 | NA |

The ranking of barriers to a circular economy, using median and mode, are presented in Table 5. The largest barrier to the circular economy is reported to be high costs, with a median and mode of 4, followed by land use with a median of 4. A value of 4 represents a large barrier according to the 5 point Likert scale. All other values are 3 (medium barrier). The largest obstacles to port industrial development is land use with a median and mode of 4, followed by high costs with a median of 4. All other barriers record medians and modes of 3, which correspondents to a medium obstacle. The largest obstacles to waterfront development, shown in Table 6, are the high costs, with a median and mode of 4, followed by land use, with a median of 4. Technology has the lowest median value of 2.5. The largest barriers to maritime clusters are high costs, with a median of 4, and city and municipal authorities with a median of 4. Pressure groups and technology report modes of 1 and medians of 2.5 and 2, respectively.

**Table 5.** Ranking of barriers to industrial development, maritime clusters, circular economy, and waterfront development.

| Barrier | Industrial Development | | | Maritime Clusters | | | Circular Economy | | | Waterfront Development | | |
|---|---|---|---|---|---|---|---|---|---|---|---|---|
| | Median | Mode | N | Median | Mode | N | Median | Mode | N | Median | Mode | N |
| Land use | 4 | 4 | 44 | 3 | 3 | 41 | 4 | 3 | 41 | 4 | 3 | 42 |
| High costs | 4 | 3 | 47 | 4 | 3 | 46 | 4 | 4 | 42 | 4 | 4 | 44 |
| Pressure groups | 3 | 3 | 44 | 2.5 | 1 | 40 | 3 | 3 | 40 | 3 | 3 | 42 |
| Private stakeholders | 3 | 3 | 44 | 3 | 3 | 44 | 3 | 3 | 40 | 3 | 3 | 43 |
| City and municipal authorities | 3 | 3 | 44 | 3 | 4 | 44 | 3 | 3 | 39 | 3 | 3 | 42 |
| Legislation | 3 | 3 | 44 | 3 | 3 | 40 | 3 | 3 | 40 | 3 | 3 | 42 |
| Technology | 3 | 3 | 44 | 2 | 1 | 39 | 3 | 3 | 40 | 2.5 | 3 | 40 |

**Table 6.** Correlation between port size and presence of a circular economy.

| Statistics | Presence of Circular Economy (Yes/No) and Port Size (Southampton System port grouping) |
|---|---|
| Correlation Coefficient (Spearman's Rank) | 0.321 |
| Sig (2 tailed) | 0.046 |
| N | 39 |

The number of respondents listing each size of barrier to the circular economy is shown in Figure 3. High costs and land use are identified as the most significant sources of very large obstacles by respondents, with all other barriers being ranked as medium obstacles by the largest number of respondents.

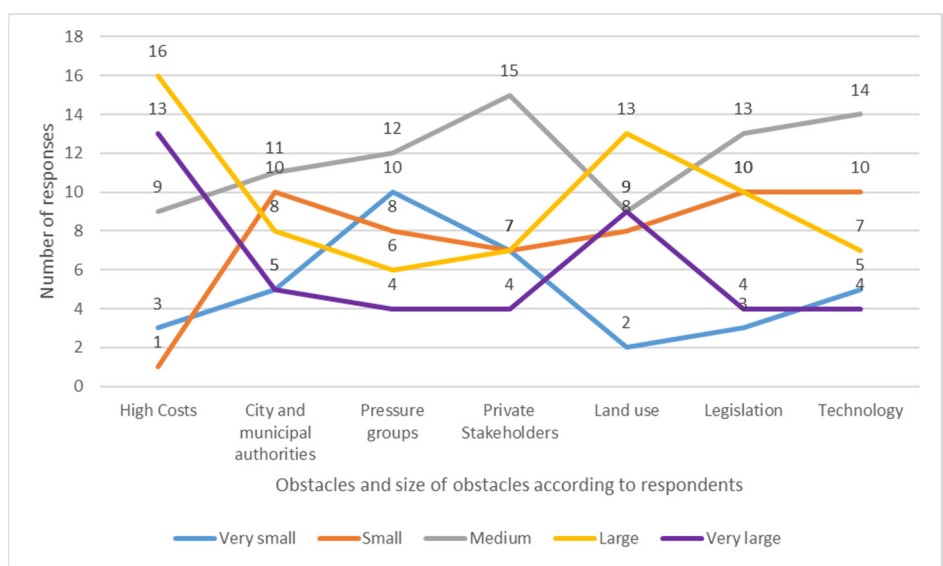

**Figure 3.** Size of obstacles to the circular economy according to respondents.

The number of respondents listing each size of barrier for industrial development is shown in Figure 4. Land use and high costs have the highest amount of respondents identifying them as very large obstacles. Technology received the highest number of respondents, listing it as a very small or small obstacle.

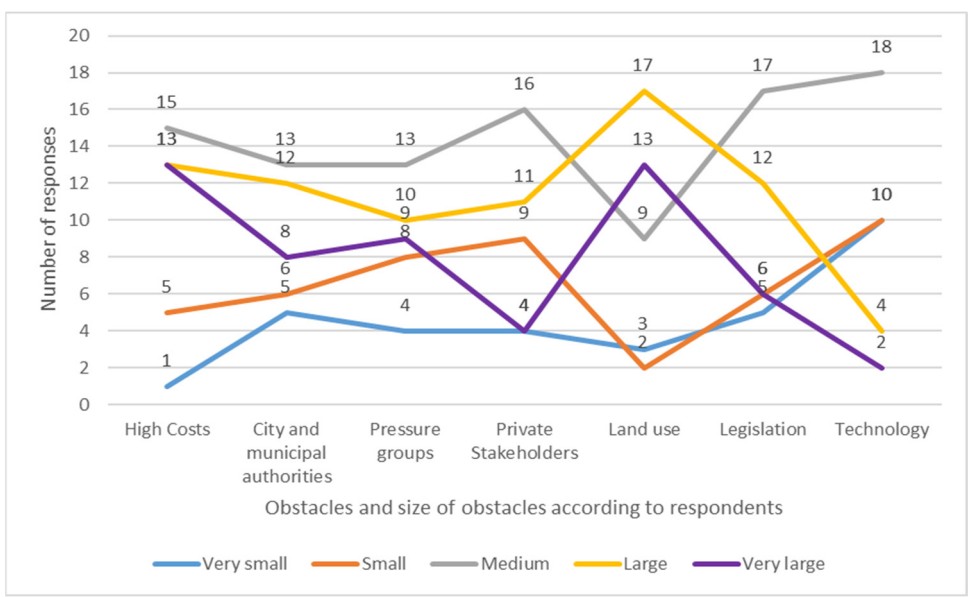

**Figure 4.** Size of obstacles to port industrial development according to respondents.

The number of respondents listing each size of barrier to waterfront development is shown in Figure 5. The barriers most likely to be rated as very large for waterfront development are high costs and land use. Technology was considered very small by a large number of respondents (15). All of the barriers, except for high costs, have medium as the most common response.

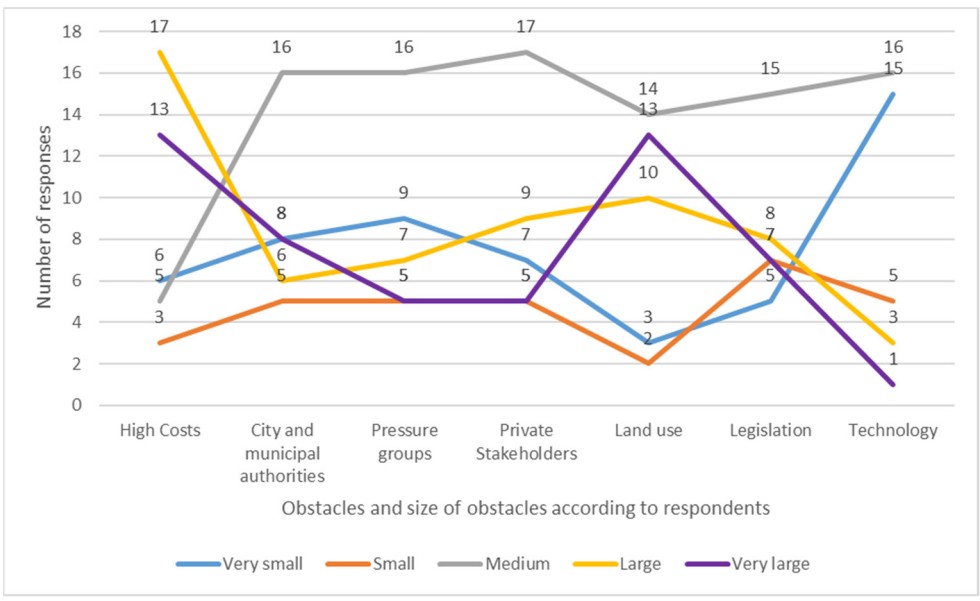

**Figure 5.** Size of obstacles to waterfront development according to respondents.

The number of respondents listing each size of the barrier to maritime clusters is shown in Figure 6. High costs and land use were rated as very large obstacles by most respondents, whilst technology was considered a very small obstacle by a large number of respondents.

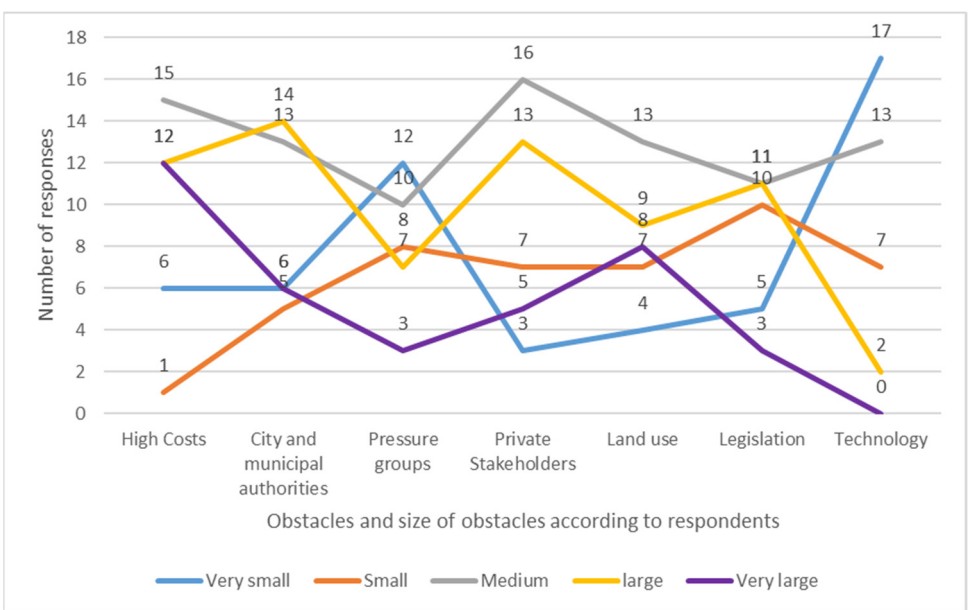

**Figure 6.** Size of obstacles to maritime clusters according to respondents.

Table 6 shows a weak positive correlation between the presence of circular economy activities in the port and port size, statistically significant at the 95% confidence level.

Table 7 shows a moderate association between the presence of a port's industrial development and respondents believing that the local population is aware of the benefits the port provides.

**Table 7.** The association between port industrial development and perceived awareness of the local population (about the benefits of the port).

| | | Response to the Question "Do You Feel the Local Population Is Aware of the Benefits the Port Provides?" | | |
|---|---|---|---|---|
| | | No | Yes | Total |
| Does the port contain industrial development? | No | 9 | 1 | 10 |
| | Yes | 9 | 13 | 22 |
| Total | | 18 | 14 | 32 |
| Chi-squared | | | | 6.73 |
| Continuity correction | | | | 4.88 |
| Significance (continuity correction) | | | | 0.02 |
| Significance (two-tailed) using Fisher's exact test | | | | 0.019 |
| Phi (strength of association) | | | | 0.46 |

The association between attitudes towards encouraging waterfront development and perceived awareness of the local population of the benefits of the port are presented in Table 8. It shows a moderate association between the desire to encourage waterfront development and the respondent feeling the local population are aware of the benefits the port provides. This result is statistically significant using chi-squared; however, when continuity correction is used, it ceases to be so.

**Table 8.** Attitudes towards encouraging waterfront development and perceived awareness of the local population (about the benefits of the port).

| | | Response to "Do You Feel the Local Population Is Aware of the Benefits the Port Provides?" | | |
|---|---|---|---|---|
| | | No | Yes | Total |
| Is the port interested in encouraging waterfront development? | No | 12 | 4 | 16 |
| | Yes | 6 | 10 | 16 |
| Total | | 18 | 14 | 32 |
| Chi-squared | | | | 4.57 |
| Continuity correction | | | | 3.17 |
| Significance (two-tailed) | | | | 0.033 |
| Significance (continuity correction) | | | | 0.07 |
| Phi (strength of association) | | | | 0.38 |

Table 9 shows a moderate strength of association between the presence of the circular economy and the respondent feeling that the local population is aware of the benefits the port provides.

**Table 9.** Association between circular economy and perceived awareness of the local population (about the benefits of the port).

| | | Response to "Do You Feel the Local Population Is Aware of the Benefits the Port Provides? | | |
|---|---|---|---|---|
| | | No | Yes | Total |
| Does the port contain circular economy? | No | 14 | 5 | 19 |
| | Yes | 4 | 9 | 13 |
| Total | | 18 | 14 | 32 |
| Chi-squared | | | | 5.78 |
| Continuity correction | | | | 4.16 |
| Significance (two-tailed) | | | | 0.016 |
| Significance (continuity correction) | | | | 0.04 |
| Phi (strength of association) | | | | 0.43 |

Table 10 shows a large strength of association between waterfront development and the respondent feeling the local population had a positive attitude towards the port. Continuity correction was not possible for this table due to having zero in one of the cells.

No statistically significant association between any of the other statements listed in the methods and the measures for improving local economic benefits were found.

A new three-step framework for encouraging circular economy activities in port cities is presented below in Figure 7.

**Table 10.** Association between waterfront development and perceived attitudes of the local population towards the port.

| | | Response to "Do You Feel the Attitude of the Local Population towards the Port Is Positive?" | | |
|---|---|---|---|---|
| | | No | Yes | Total |
| Does the port contain waterfront development? | No | 9 | 0 | 9 |
| | Yes | 11 | 12 | 23 |
| Total | | 20 | 12 | 32 |
| Significance (two-tailed) using Fisher's exact test. | | | | 0.01 |
| Phi (strength of association) | | | | 0.53 |

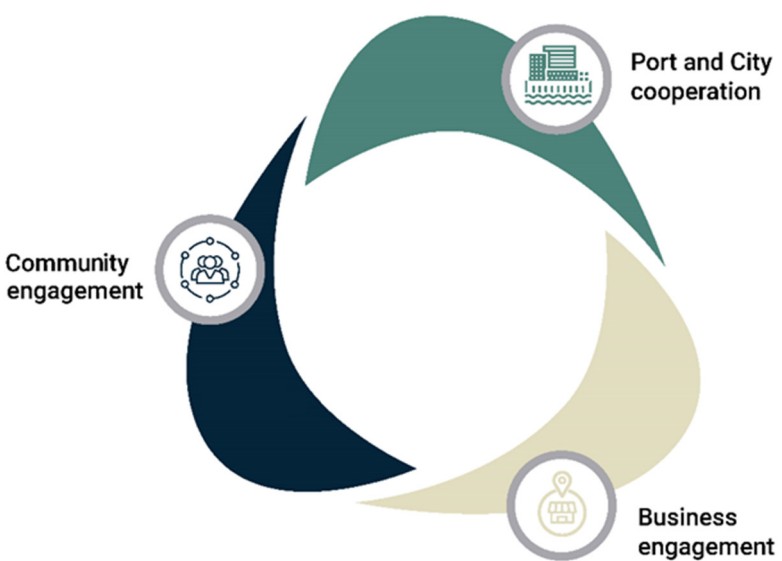

**Figure 7.** Framework for increasing circular economy activities in port cities.

Step 1: Port and city cooperation:

- Annual summit between port and city stakeholders to approximate the current levels of a circular economy within the port city, progress so far, key opportunities, and the potential for the future, key businesses/industries, and a plan of action for the following year. This summit could be brokered by a third party, such as relevant consultancies or universities, and bring together port and city authorities, as well as key port-related stakeholders.
- A steering committee containing representatives of key stakeholders should be established to oversee the subsequent steps.
- Key indicators for each port city can then be targeted by the stakeholders to provide a form of assessment, such as those suggested by Girard and Nocca [84].

Step 2: Business engagement:

- Hold an open day intended to raise awareness of the opportunities and attract potentially interested businesses within the port city. Cooperative businesses in the port city can be awarded "circular economy champions" status.
- To be awarded circular economy champion status, businesses must commit to a plan of action to increase circular economy activity within their business with actionable targets.
- Efforts should be made to establish if there are any ways for local businesses to make use of the waste created by other businesses in the port city. This can be done by participating businesses creating an inventory of the waste, assets or end of life

equipment they are expected to produce that other businesses and interested parties can access. This will then create a port city circular marketplace, as well as potentially inspiring further business opportunities for interested parties, for example, the repair and reuse of end of life fishing gear.

Step 3: Community engagement:

- Inspire the wider community within the port city by engaging with schools, community centres, higher education, the media, etc. Raising awareness of the benefits of circular economy thinking and any success stories within the port city. This can make use of existing port events, port centres, educational activities, etc.

## 4. Discussion

Many of the ports in the study have already adopted one or more methods for increasing local economic benefits. Among these, a waterfront economy is the most widely adopted, followed by industrial development. There is an association between adopting one method and adopting all of the other methods, suggesting that ports contain a mix of these rather than being specialised. The presence of maritime clusters is most strongly associated with the presence of circular economy with a Phi of 0.45. However, there is a moderate strength of association with industrial development (0.37) and waterfront development (0.34). Port industrial development shows the strongest association with waterfront development (0.49), which may be influenced by the fact that a high level of industry may suggest the port has grown beyond the traditional port area close to the city centre, leaving older areas of the port free for redevelopment. A circular economy shows higher levels of association, with moderate associations with maritime cluster presence (0.45) and industrial development (0.43), and a high association with waterfront development (0.5). The associations between these methods shows that ports regularly have a mix of maritime clusters, industrial development, circular economy, and waterfront development. It may also suggest that some of the methods are complimentary, with industrial development providing components of a maritime cluster, and circular economy making use of waste products created by industrial development, for example. It also suggests that there are active ports that are likely to engage with multiples of these measures, and inactive ports that are not engaging with any of them. It is therefore important for further work to identify the reasons why ports become active, and find ways to help ports begin the process of implementing ways to increase their local economic benefits, which may help encourage greater levels of circular economy. All of these measures appear to have the potential to compliment the circular economy.

A circular economy sees a large increase of interest compared with current levels (59%) in Table 1. This suggests that, as well as being ideally placed to be at the centre of the cultural paradigm shift to a circular economy, called for by Gravagnuolo et al. [31], ports have a clear interest in embracing this transition globally and across all sizes in this study. Table 5 shows that circular economy in ports faces similar barriers to industrial development and waterfront development, with the largest obstacles being high costs and land use. Figure 3 shows that high costs and land use have the highest frequencies of being identified by respondents as very large obstacles to circular economy. This supports the findings of Mont et al. [29] and Van den Berghe et al. [28] on general CE barriers, as well as the findings of Carpenter and Lozano [85], who stated that economic viability is the first requirement for CE in ports. Technological limitations are not shown to be a significant source of large or very large obstacles to a circular economy. Addressing the key obstacles of high costs and land use will be essential if ports are to achieve their full potential within the circular economy, and these are both obstacles that the port may be able to work with city authorities and private stakeholders to address, as Figure 3 shows, they are not a major source of obstacles. Given the considerable benefits a circular economy can provide for cities, encouraging a circular economy in ports may provide a triple win scenario by benefitting both ports and cities, as well as the wider society. This may be encouraged by existing programs, such as the Scottish government's proposed green

ports [86], which commits ports to supporting sustainable and inclusive growth in local areas, whilst transitioning to net zero emissions, or the European environmental initiative EcoPorts [87]. It is clear from the data that there is a desire among ports to increase circular economy activities, and that the barriers investigated in this study are comparable in size with those for other measures for increasing economic benefits. This suggests that a lack of circular economy adoption may therefore be due to cultural and organizational barriers as stated by Kirchherr et al. [24], such as a lack of cooperation, rather than financial or technological barriers. Therefore, measures to overcome these cultural barriers are needed.

Support for maritime clusters remains consistent with current levels (Table 1). High costs are reported to be the biggest obstacle, with private stakeholders and city and municipal authorities also providing a source of large obstacles (Figure 6). This may be because a maritime cluster is a network of inter-related industries, cooperation with city authorities, and private stakeholders is essential for their development. The key sources of barriers are different for maritime clusters when compared with the other measures, with, for example, land use being a smaller obstacle. This may be because some of the industries required to form a maritime cluster may require land within the city itself rather than on port land, making it an issue the port has less influence over. It may be possible to create maritime clusters focused around the circular economy, and this may be a more suitable way for ports to increase circular economy activities if land use is an issue, or in cities that lack large amounts of industrial development.

There is a considerable fall in interest in waterfront economy when compared with current levels. This may simply be because waterfront development has often taken place in disused or abandoned port land. It often takes place when a port—or a specific area of land used by the port—has entered a decline and is available for new uses, which is often an externality of port operations as discussed by Hoyle [60] and Saz-Salazar et al. [1]. It may be the case that potential development opportunities in this area have already been exhausted, as present levels of adoption are shown to be high. This may explain why port authorities are not necessarily highly interested in encouraging it in the future. Figure 5 and Table 5 show that barriers to waterfront development are not higher than the other methods. This suggests that port authorities are simply less interested in this option, rather than facing too many obstacles to pursue it. It may be something that port authorities feel is more under the control of the cities themselves.

Industrial development is the most widely adopted option and is the most desired option for the future (Table 1). It is therefore essential that this development be linked to the circular economy in order to mitigate the negative impacts. It is likely due to these impacts that this option faces very large obstacles (Figure 4). High costs and land use are reported to be the largest obstacles; and pressure groups, city and municipal authorities, and legislation were shown to be sources of considerable obstacles. This shows that industrial development faces greater opposition and scrutiny from wider society, highlighting the growing awareness of the negative impacts industrial development can create, and the scrutiny placed upon it, as a consequence. Utilising the circular economy may be an effective way to allow industrial development within port areas, whilst reducing the negative impacts, creating additional benefits, and decreasing the barriers posed by pressure groups and city and municipal authorities.

Table 4 shows that interest in encouraging the various methods are also associated with positive attitudes towards other methods. The moderate strength of association between encouraging industrial development and encouraging circular economy may be due to the potential symbiotic nature of these industries. Maritime clusters may be interlinked with industrial development and the circular economy, as these may add extra components to the maritime cluster. In the future, industrial development should be undertaken alongside a circular economy in order to reduce the negative impacts of this development. The association concerning the desire to encourage both of these is, therefore, potentially an encouraging sign that this may be the case.

Table 6 shows there is a weak, but statistically significant, correlation between the presence of a circular economy and port size. This indicates that ports are more likely to implement a circular economy when they have greater volumes of traffic and passengers and, therefore, have a greater pool of resources, waste products, facilities, and potential consumers. This suggests that smaller ports may be less likely to implement a circular economy due to this, highlighting the challenges smaller ports face, discussed in the literature [10]. Further research should therefore be undertaken to encourage greater adoption of a circular economy in smaller ports.

There is a high level of association between encouraging waterfront development and encouraging industrial development, with a Phi of 0.57. This is the strongest level of association in Table 4. This may be the case, due to the fact that old port areas can be redeveloped into waterfront areas, as outlined by Hoyle [60], as new port areas are made available. This may allow ports to pursue the industrial development they desire whilst providing something that benefits the city and local residents. This high level of association suggests that ports may be willing to move some of their detrimental activities away from the waterfront to free this up for waterfront development. This is a continuation of the process of obsolete port areas being renewed via waterfront development, as described by Hoyle [60]. Access to the waterfront can provide numerous benefits to cities and for this reason, is very desirable, whereas industrial development may face more resistance. This is illustrated in Figure 4, which shows that city and municipal authorities are more likely to provide a large obstacle to industrial development than they are for waterfront development and a circular economy.

There is a moderate association between the perceived awareness of the local population towards the benefits the ports provide and the presence of industrial development and a circular economy, and an interest in encouraging waterfront development (Tables 7–9). This suggests that local residents are more likely to feel the benefits provided by industrial development and a circular economy than those provided by maritime clusters and waterfront development. Waterfront development may not be perceived as a benefit provided by the port directly, and the public may not be aware of the existence of maritime clusters if they are not directly involved in industries relating to the port. Waterfront development has a high level of association with perceived positive attitudes towards the port (Table 10). This suggests that waterfront development, rather than being perceived as a benefit provided by the port, improves attitudes towards the port in a more general sense. Overall, this shows that maritime clusters are, potentially, a method that produces the least palpable benefits for local residents in terms of how it impacts their attitudes towards the port and its benefits.

The data shows high levels of support for circular economy and highlights that high costs and technological limitations are not greater barriers than they are for other already widely adopted measures. This suggests that low levels of CE adoption may be due to cultural and organisational factors. The framework provided in Figure 7 may help address some of the cultural and organizational challenges in port cities. Step 1 seeks to create the enabling factors of networking, cooperation, and involvement of stakeholders [10,20,31,47], as well as establishing leadership from the top. Step 2 encourages knowledge transfer, cooperation, and incentivisation by awarding circular economy champion status, all of which have been identified as important in the literature. Step 3 addresses the often-neglected social dimension of CE, whilst providing added recognition for participants, increasing incentivisation and helping move CE away from being a niche discussion, and of interest to wider society. This framework can provide an annual cycle of improving circular economy performance in port cities and allows local stakeholders to control and influence the process, enabling it to be tailored to the specific needs of the port city.

Further work is needed to elaborate on the barriers faced by ports in relation to the CE, as this questionnaire had closed answers, which may limit the findings, and not capture the full range of responses. Future work should also attempt to assess the effectiveness of frameworks, such as the one proposed in this paper. Adopting circular

economy principles may pose significant challenges to ports due to their existing business models [37]. Creating a circular economy may therefore require a paradigm shift in port business models to accommodate this change. This may be achieved through various means, such as the incentivised return of used goods for reuse and repair, the sharing of assets and resources between businesses and key stakeholders via the sharing economy, or asset management to plan the reuse, repair, or redeployment of assets [88], amongst others. Future work is needed to create suitable business models to allow ports to fully embrace the circular economy.

### 5. Conclusions

This study highlights the clear potential for greater adoption of a circular economy within ports. It is the first time port industrial development, maritime clusters, waterfront development, and circular economy have been investigated together across such a large sample of countries (26), across multiple continents, giving the findings a global view that has been absent in previous work, which mostly focused on case studies from individual countries or regions. The potential and willingness of ports to be on the frontline of the transition to a circular economy, globally, was clearly identified for the first time. There are numerous circular economy opportunities in ports, such as end-of-life fishing gear, residual heat capture from vessels, end-of-life vessels, scrubber sludge and wastewater, industrial waste, and scrap metals. Current levels of a circular economy are low, and can be expected to see considerable growth in the future if ports are able to overcome the barriers to the implementations they face. The barriers to a circular economy are similar to those faced when ports pursue industrial development and a waterfront economy, mainly relating to land use and high costs, which are obstacles that have been overcome before, and obstacles that cities could work together with ports to overcome. Considering the significant economic, social, and environmental benefits a circular economy can provide, the circular economy should be encouraged in port cities as a win-win scenario for both parties. Port authorities are interested in encouraging greater levels of a circular economy in the future, and city authorities can benefit significantly from this. Port cities should therefore be well placed to embrace the transition to a circular economy. If port cities aim to be more sustainable in the future, circular economy activities may provide a good balance of the three pillars of sustainable development (environment, economy, and society). City authorities, regional and national governments, and supranational bodies, such as the European Union, should therefore be encouraged to assist ports in implementing the circular economy, especially if they seek to alleviate the negative impacts of any potential industrial development the ports may wish to encourage. Greater appreciation of the benefits of ports is perceived among the local population if they have adopted a circular economy and industrial development activities. Ports seeking to create goodwill amongst the local population towards the port should consider circular economy activities as a more sustainable way to improve local attitudes. Further work is needed to identify the views of city authorities towards a circular economy, to more clearly identify ways for cooperation between ports and cities.

Cities may be able to cooperate with ports in moving industrial development away from the city centre, reducing the negative impacts, and freeing up land for waterfront development, continuing the process identified by Hoyle [60]. The high level of association between interest in waterfront development and industrial development from ports could be used as a way to balance the competing interests of ports and cities, freeing up access to the waterfront whilst allowing industrial development to take place elsewhere. A circular economy may be the best way to enable industrial development whilst reducing the negative impacts it creates, overcoming some of the barriers this type of development faces, allowing port cities to obtain the benefits.

If ports are to add greater economic benefits, then the key obstacles of high costs and land use need to be addressed. This may be achieved through greater cooperation among ports, cities, and relevant stakeholders, such as joint planning. Maritime clusters and

industrial development remain desirable options for ports; however, of the four options considered in this paper, a circular economy appears to be the most well placed as a strategy, with potential for growth that can provide these benefits, attract support from local residents, and allow ports to flourish, whilst embracing a greater level of sustainability. The framework provided by this paper creates a foundation in which this process can begin.

**Author Contributions:** Conceptualization, T.R., I.W., J.P. and N.C.; data curation, T.R.; formal analysis, T.R.; funding acquisition, I.W., J.P. and N.C.; investigation, T.R.; methodology, T.R., I.W., J.P. and N.C.; project administration, I.W. and J.P.; Supervision, I.W., J.P., N.C. and M.O.; visualization, T.R.; writing—Original draft, T.R.; writing—Review and editing, T.R., I.W., J.P., N.C., M.O. and S.O. All authors have read and agreed to the published version of the manuscript.

**Funding:** The authors are pleased to acknowledge that this research study was partially funded by the EPSRC Centre for Doctoral Training in Sustainable Infrastructure Systems (EP/L01582X/1) and partially funded by Ramboll.

**Institutional Review Board Statement:** The study was conducted according to the guidelines of the Declaration of Helsinki, and approved by the Institutional Review Board (or Ethics Committee) of The University of Southampton (ID 55821 and 16.02.2020).

**Informed Consent Statement:** Informed consent was obtained from all subjects involved in the study.

**Data Availability Statement:** All subjects gave their informed consent for inclusion before they participated in the study. The study was conducted in accordance with the Declaration of Helsinki, and the protocol was approved by the Ethics Committee of the University of Southampton's Ethical approval process, with Ethical approval ID number 55821.

**Acknowledgments:** We would like to acknowledge the contribution of the peer reviewers in helping to improve this paper.

**Conflicts of Interest:** This work was in collaboration with engineering consultancy Ramboll, who are the industrial sponsors of this research, in collaboration with the UK's Engineering and Physical Sciences Research Council (EPSRC). This research may help Ramboll refine their service offerings in the future; however, this had no impact on the academic integrity of the work, which has been approved by the University of Southampton's ethical approval process.

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
