# Peer review of "A Virtuous Circle? Increasing Local Benefits from Ports by Adopting Circular Economy Principles"

_sustainability, doi:10.3390/su13137079_

Round 1

Reviewer 1 Report

Ports is a complex sector with a high economic performance. This research could be improved with ports clusters and its impact on economy. There are a few studies about it in Portugal; another recomendation, it can be presented the economics linkage with other sectors and its impact on economic and social performance. In my opinion, the staments about the circular economy are very poor and need to be improved. 

Reviewer 2 Report

I congratulate the authors for their very interesting project and well written paper.

I see substantial potential in the paper to make contribution to sustainability research. However in its current form the contribution to knowledge in CE economy and sustainability is very limited.

The way the paper is organized and results discussed is aligned with a transportation or shipping journal but not with a sustainability journal. For a sustainability journal, the paper needs to emphasize how the research question aligns with a gap in our knowledge about CE ( since that is the central sustainability concepts here). So, I recommend major revisions to re-write the paper to better justify the relevance and contribution of your research to CE.

  1. Introduction: The introduction does not even mention circular economy. After the abstract CE is not mentioned again until page 88. You need to re-arrange the intro so it focus on CE and explains to a sustainability reader why your research is important to advance knowledge in CE. There are many recent literature reviews in CE. I recommend you to find lit. reviews published in 2019-2021 and look at the gaps in CE knowledge and areas for further research identified in the discussion/conclusion of this paper and see how your paper can contribute to any of the gaps or lines of research there mentioned. Then you can look at  papers already published in CE in ports to identify how you contribute to this specific literature. I just entered “ports and CE” in Google Scholar and found 15 relevant papers in the last 3 years, including 2 papers published in 2020 in Sustainability.  

 Haezendonck, E., & Van den Berghe, K. (2020). Patterns of Circular Transition: What Is the Circular Economy Maturity of Belgian Ports?. Sustainability, 12(21), 9269.

Mańkowska, M., Kotowska, I., & Pluciński, M. (2020). Seaports as nodal points of circular supply chains: Opportunities and challenges for secondary ports. Sustainability, 12(9), 3926.

  1. The introduction should define CE and explain what do you mean by CE principles. CE principles are in your title and abstract but there are never defined in the body of the article.

You mentioned that there are more than 140 definitions of CE. The one you chose does not capture dominant views and for me it is not very clear.  I would recommend you to use more widely endorsed definitions, for instance from Ellen McArthur foundation, or the European Commission, or the definition recommended by Kircherr , or definitions in other papers that look at CE concept overlapping with other concepts ( e. Senehm et al, 2019; 2021, Henry et al, 2021)

  1. Literature Review: Your questionnaire has 4 main concepts: maritime cluster, port industrial development, maritime cluster and Industrial economy. The focus of your paper and review should be circular economy, with the other 3 themes substantially shortened and linkages or potential linkages for CE highlighted, This could be simply done in a thematic framework where you articulate graphically why you study this 4 themes together and what relations you expect to find ( trade-offs/ tensions). In the review of CE, you could expand on the different definitions (A table showing definitions of CE in lit.review in the last 3-5 years would help.) then describe what are the CE principles, the benefits of CE identified, critiques and the areas for further research; since there are many sist. Literature reviews on the topic, you should comment on what they say.  The next section should be about CE barriers ( as this is one the main aspects in you survey). Again, these is plenty of literature here, so we need to know what such literature says, specially prior literature reviews. The last section should be about literature in CE and ports, where you include the examples and sources you already have, plus several you do not have. In this section, you then discuss only the barriers for CE identified in the CE and ports literature. In your current discussion of barriers for CE and ports, the majority of sources you cite do not specifically discuss ports. They are papers about CE barriers in general. While it is not a problem for you to propose that these barriers apply to ports as well (in this case your survey will be providing empirical evidence for what you proposed, and that is a contribution) , the way you present it leads to assume they have been specifically found to be barriers for ports. It may be the case, that you just cited the wrong references, and these barriers are mentioned in the literature about CE and ports, in such case please revise.
  2. Your paper describe relations between constructs but it is not driven by theory. While I do not expect you to make a theoretical contribution I recommend that you propose some framework or model to increase circularity in ports, building on the relations identified in the analysis.
  3. The discussion should have : a) more focus on E, b) more discussion of how your results compare with the literature in CE, barriers for CE and CE in ports; and how your findings extend these discussion; including limitations and areas for further research in CE and ports.
  4. Good luck!

Round 2

Reviewer 2 Report

Congratulations and thanks you very much for the substantial effort you have taken to foreground the contribution of the paper to sustainability and Circular Economy. I am glad you found the peer review helpful, I enjoyed reading your paper and learning about your findings. All the best 

Author Response

Thank you very much for your comments and contributions in improving this paper, I am pleased you are satisfied with the changes!